# Effectiveness of an Unguided Online Intervention for Sexual Pleasure in Women: A Randomized Controlled Pilot Study

Michèle Borgmann [1,*], Lucca Michelle Brandner [1], Linda Affolter [1], Julia Vonesch [2] and Stefanie Gonin-Spahni [1]

1   Department of Health Psychology and Behavioral Medicine, Institute of Psychology, Faculty of Human Science, University of Bern, 3012 Berne, Switzerland
2   Applied Social and Health Psychology, Department of Psychology, Faculty of Philosophy, University of Zurich, 8050 Zurich, Switzerland
*   Correspondence: michele.borgmann@unibe.ch; Tel.: +41-31-631-56-38

**Abstract:** The importance of sexual pleasure as a factor promoting sexual and public health is increasingly recognized. Nevertheless, hardly any theory-based and empirically tested interventions exist for fostering sexual pleasure. Consequently, we developed an unguided online intervention called PleaSure to promote sexual pleasure in women. In a randomized controlled pilot trial with a mixed-method design, we evaluated the effectiveness of PleaSure by comparing the intervention group to a waitlist control group in pre–post measurements over 4 weeks. With 657 participants ($M_{age}$ = 31.46, $SD_{age}$ = 8.78), we evaluated an index of sexual pleasure and five facets: sensual pleasure, pleasure-related mastery, pleasure-related validation, interaction pleasure, and bonding pleasure. The results show that the online intervention primarily strengthened the intrapersonal domain of sexual pleasure by increasing pleasure-related mastery. Neither the other facets nor the index was significantly influenced by the intervention. Although the effects of the quantitative data are small, the qualitative data support overall positive effects on participants' sexual experience. We discuss the content of the intervention and the methods used. Our pilot study suggests that sexual pleasure can be promoted but that major improvements are needed to the intervention's content and design to do so effectively. Therefore, future studies are encouraged to further develop and implement such resource-efficient interventions in clinical and non-clinical samples to better understand the importance of sexual pleasure to sexual health.

**Keywords:** sexual pleasure; sexual health; online intervention; randomized controlled trial

## 1. Introduction

Sexual pleasure can be defined as "physical and/or psychological satisfaction and enjoyment derived from solitary or shared erotic experiences, including thoughts, dreams and autoeroticism" [1], and it constitutes one of the main reasons for engaging in sexual situations and becoming sexually active [2–5]. Sexual pleasure describes the positive feeling that occurs when rewards are expected, attained, and enjoyed during sexual activities [6]. Because the rewards achieved through sexual activity can be diverse, sexual pleasure is seen as a multidimensional construct [7] and can be divided into three domains: hedonic, intrapersonal, and interpersonal. These domains in turn comprise facets: sensual pleasure in the hedonic domain, pleasure-related mastery and validation in the intrapersonal domain, and interaction and bonding pleasure in the interpersonal domain. Each facet is related to a possible reward that can be derived from sexual activity [6].

Sexual pleasure has been considered an essential component of sexual health and sexual rights for several years [8–12]. This is supported by a growing body of evidence showing positive associations between sexual pleasure, sexual health, and health-related outcomes [13]. A review found that sexual satisfaction, pleasure, and positive self-esteem

have a positive impact on sexual health as well as mental and physical health [14]. The latter can also be supported by an older study showing that enjoyment of sexual activity is associated with longevity in women [15]. However, results from research on sexual pleasure are also promising in terms of psychological and behavioral outcomes [16]. On the one hand, studies have shown that sexual pleasure is positively correlated with autonomy, self-esteem, and empathy [17,18]. On the other hand, a recent study on sexual behaviors has shown that sexual pleasure is related to several health indicators, such as communication about sexually transmitted infections (STIs), condom use, and the absence of sexual problems [19]. Recently, sexual pleasure was even declared one of four pillars of a comprehensive public health approach to sexuality [12].

These findings highlight the importance of sexual pleasure to overall well-being and the need for interventions focused on increasing sexual pleasure. Therefore, the World Association of Sexual Health (WAS, 2019) highlights in the Declaration on Sexual Health that sexual pleasure should be accepted and supported by acknowledging "sexual pleasure as a component of holistic health and well-being: The right to sexual pleasure should be universally recognized and promoted" [20].

### 1.1. Sexual Pleasure in Sexual Health Interventions

Many sex-positive interventions and projects have been initiated and evaluated over the past two decades, resulting in increased recognition of and focus on positive sexual health constructs such as sexual pleasure. Research indicates that interventions that incorporate sexual pleasure can lead to improvements in sexual health knowledge and attitudes and behaviors such as learning how to communicate with partners and practicing safer sex [21–24]. This was last confirmed by a recent systematic review that examined 33 studies targeting HIV/STI risk reduction through a pleasure-based approach. The authors found that promoting sexual pleasure leads to a reduction in risk behaviors associated with sexual health [25]. Other reviews and meta-analyses show congruent findings in which pleasure-based interventions reduce sexual risk-taking and improve sexual health [21,22,24,26]. The repeated demonstration of the importance of sexual pleasure to public and sexual health has provided a foundation for a pleasure-based approach to sexual health and sexual rights.

However, in previous studies, sexual pleasure has been studied as a predictor of sexual health rather than an outcome [21–25]. Hardly any studies have examined the effectiveness of interventions for increasing sexual pleasure in the general population.

### 1.2. Online Sexual Health Interventions

In recent decades, a promising trend in sexual health interventions has emerged in which interventions are increasingly delivered online. The effectiveness of online interventions for sexual health has already been confirmed several times. For example, online interventions have focused on (1) various sexual problems and dysfunctions [27–34], (2) STIs [35,36], and (3) sex education [37–40]. Online interventions offer many advantages: they allow easy access, are flexibly available, can be conducted anonymously, save time and travel, and are very cost-effective [41–43]. Indeed the European Society of Sexual Medicine has recently clarified that online sexual health interventions can offer fundamental opportunities to improve sexual health in the general population [44].

However, the online interventions conducted so far and the interventions involving sexual pleasure mentioned above have focused primarily on at-risk or clinical subgroups. To the best of our knowledge, only one observational study has tested the effectiveness of a website (OMGyes.com, accessed on 1 March 2021) that presents masturbation strategies as a resource for empowering women to enhance their understanding, support, and enjoyment of sexual pleasure [45]. The participants were asked to explore OMGyes.com over a 4-week period. No further instructions were provided on frequency or approach to resource use. Using the OMGyes.com website for 4 weeks had a positive impact on how women thought and felt about their sexual pleasure and how they understood and communicated their own preferences to their partners [45]. By targeting the general population, defining sexual

pleasure as an outcome, and using the online setting, the study summarizes what has been lacking so far. The study provides important evidence that sexual pleasure can be targeted, but it has three main limitations: first, the study included no control group; second, it was not conducted as a randomized controlled pilot trial (RCT); and third, the intervention is not theory based.

*1.3. Sexual Pleasure and Sexocorporel*

The Sexocorporel approach can provide a theoretical background for interventions for promoting sexual pleasure because this is the central goal of Sexocorporel sexual therapy [46]. Sexocorporel is a comprehensive view of human sexuality that considers the physiological, emotional, cognitive, and relational components involved in a sexual experience [46–48]. According to Sexocorporel, sexual pleasure depends on individual knowledge and learning processes; thus, psychoeducational elements and self-experience elements are central to promoting sexual pleasure. Such practical elements of Sexocorporel include exercises and reflections that provide resources to experience a positive body image and genital self-image. To achieve this, they focus primarily on self-stimulation and individuals' new experiences with their own bodies through conscious changes following the three dimensions of body movement, rhythm, and muscle tone during sexual activity [47,48]. Movement of the pelvis and alternating phases of contraction and relaxation of the pelvic floor are associated with greater pleasure and orgasms during sex [49]. By learning and mastering a variety of arousal modes through mindfulness, body self-exploration, and concrete exercises to increase and intensify arousal, individuals can expand the spectrum of their sexual pleasure [47–49]. Therefore, Sexocorporel may provide a theoretical framework for an intervention for promoting sexual pleasure.

One study has already empirically tested the Sexocorporel approach in a face-to-face therapy for men with premature ejaculation and shown promising results for sexual function and sexual satisfaction [50]. In addition, three German-language manuals written by sex therapists discuss the basics, methods, and use of Sexocorporel [51–53].

*1.4. The Current Paper*

We drew on the theoretical background of the Sexocorporel approach to develop a 4-week unguided online intervention called PleaSure. PleaSure is the first online intervention based on Sexocorporel and incorporates psychoeducational elements and specific exercises for promoting women's sexual pleasure. Our decision to prioritize research on women's sexual pleasure before men's was motivated by the historical imbalance in which women's pleasure has been given less attention and importance compared to men's [5,54–56]. By focusing on women's pleasure, our study aims to contribute to the growing efforts to promote sexual agency and pleasure for all women. The intervention targets the general female population and uses sexual pleasure as a preventative factor that promotes sexual and public health. Thus, the intervention addresses the core of sexual health policy suggested by the European Society of Sexual Medicine [44].

The aim of this pilot study was to test the effectiveness of the online intervention PleaSure using a mixed-method design in an RCT comparing an intervention group (IG) with a waitlist control group (WCG).

Our research questions were as follows:

Research question 1 (RQ1): How do the facets of sexual pleasure change among the IG before and after the online intervention compared to the WCG? We expect that women's facets of sexual pleasure in the IG will be significantly higher in the postintervention measurement than in the preintervention measurement compared to women in the WCG.

Research question 2 (RQ2): How do the sexual pleasure facets change in the IG from pre to post to follow-up as a result of the online intervention? We expected the positive outcomes to persist at 4-week follow-up among women in the IG.

## 2. Materials and Methods

### 2.1. Description of the Online Intervention PleaSure

PleaSure is an unguided online intervention for women from the general population to promote sexual pleasure using psychoeducational elements and specific exercises derived from Sexocorporel. Thus, the intervention aims to promote sexual pleasure among a wide range of women, rather than just a specific subgroup (e.g., women with sexual dysfunctions). The content of the intervention was developed from the theories of Sexocorporel [46–48,57], the German-language Sexocorporel manuals [51–53], and an exercise manual with hands-on Sexocorporel exercises [58]. The online intervention was conducted in German and was divided into four steps, which were completed by the participants over a period of 4 weeks. Each week covered a thematic focus and specific learning outcomes that were always presented at the beginning of a new week (see Table 1). The website (www.pleasure-studie.ch, accessed on 1 December 2022) begins with some basics, providing participants with important terms and theories relevant to the Sexocorporel approach and an anatomical summary about the female reproductive organs, so that all participants have the same prior knowledge. After this, the four steps are as follows:

1.  The first week focuses on mindful awareness of the body. The knowledge section includes information on mindfulness, the parasympathetic and sympathetic nervous systems, and Jacobsen's progressive muscle relaxation (PMR); the link between body and mind is a key theme in Sexocorporel [48,52]. The exercises for this week are a mindful-based body scan and a PMR exercise [58] that treats the whole body including the genitals and can be listened to as audio or read as text.

2.  The second week focuses on exploring the vulva, as Sexocorporel assumes that the better the genital self-image is, the better the sexual experience will be [53]. Thus, general knowledge is provided about the vulva, and a summary of the OMGyes study describes various forms of genital stimulation [59]. The exercise is on genital self-image [58]. In the first part of the exercise, the vulva is explored with a hand mirror. In the second part of the exercise, the vulva is explored with the fingers.

3.  The third week focuses on the habitual arousal patterns and the variation of tension and relaxation of body regions that is integral to Sexocorporel practice [46,47]. Information was provided on arousal patterns and on the relationship between tension and relaxation. As an exercise, the participants were asked to masturbate while varying their habitual arousal patterns [51]. At the end of this step, the participants are encouraged to perform the exercise in couple sex as well. The participants receive input on the topic of consensus and communication in couple sex because it is known that experiences and skills acquired in solo sex can be transferred to couple sexuality [49].

4.  The fourth week focuses on the element movement and the body parts the pelvic floor and the inner vaginal space as these are central body regions for Sexocorporel [46,52]. The arousal modes were repeated, and reflection questions were asked about muscle tension during masturbation. The psychoeducative element relates to the pelvic floor, excessive muscle tension, and associated pain. An exercise on tensing and relaxing the pelvic floor provides a practical demonstration for the participant. In addition, the Sexocorporel double swing exercise [46] is introduced with a text and two videos. The double swing combines a movement of the pelvic swing with a movement of the chest, neck, and head [58] and is associated with greater physical and emotional intensity in sexual arousal [49]. Another exercise is the bullet fantasy journey exercise [58], which uses audio to help the participant imagine a ball making its way through her vagina.

**Table 1.** Procedure and content of the study.

| | Topic | Psychoeducative Elements | Exercises | Learning Outcomes (in This Week You . . . ) |
|---|---|---|---|---|
| Start | Basics | Important theories (embodiment, mindfulness, and Sexocorporel) and terms (gender and sexuality) Refresh anatomy knowledge (vulva, vagina, clitoris, and pelvic floor) | Reflect actual state of own sexuality Set personal goals | Reflect your current state in sexuality and set personal goals for yourself, how intensively you want to deal with the course and what you want to achieve in the further development of your sexuality. Refresh your knowledge of the anatomy of sex and perhaps learn something new about it. |
| 1st week | Perception of one's own genitals | Exploration on mindfulness and progressive muscle relaxation (PMR) | Opening exercise: exploration with the hand Mindful body scan including the genitals | Refresh your knowledge about mindfulness. Perceive your body in its present state. Explore and observe your body in a mindful, non-judgmental, and curious way. |
| 2nd week | Discovery of one's own genitals | Description of the vulva and different forms of genital stimulation | Mindful exploration of the vulva with a hand mirror Mindful exploration of the vulva with fingers | Recognize the diversity of different vulvas. Establish acceptance and a sense of pride towards your genitals. Learn about different aspects of genital touch and stimulation. |
| 3rd week | Arousal tension and relaxation | Description of sexual response cycle, orgasm, and arousal modes | Masturbation exercise Getting to know their own arousal modes | Learn theoretical background knowledge about the sexual response cycle and ways of increasing arousal. Get to know and expand your own arousal patterns through pleasant touch of the genitals and the whole body. |
| 4th week | Movement | Explanation of the pelvic floor, movement, and vagina | Tightening the pelvic floor Double swing Imagination to the inside of the vagina | Learn about the relevance and modulation of the pelvic floor. Learn that the increase in perception and arousal can be regulated by tension, relaxation, and movement of the pelvic floor. Feel the spread of sensory perception in the genitals with the help of movements of the pelvis and a fantasy journey. |
| Closing | Review | Encouraging further practice (also in couple sex) Further inspiration (links to info pages) | Review and reflection | |

At the end of each step, central references are summarized, contact information is provided, and the participants are encouraged to repeat the exercises several times during the week. At the end of the online intervention, the participants were asked some reflection questions. In addition, they were taught how to implement the contents learned in their everyday lives. Excerpts from the website can be found in the Supplemental Materials S1–S7.

PleaSure offers various advantages associated with this unguided online format. These include comparatively easy access, flexible availability, anonymity, and cost-effectiveness [41]. The online setting proved to be a particularly useful and practical tool because the RCT was conducted during the COVID-19 pandemic, when face-to-face contact was especially difficult.

### 2.2. Procedure and Study Design

A two-arm RCT with three repeated measures was conducted to compare participants who used the internet-based unguided intervention PleaSure and thus constituted the IG to those assigned to the WCG. Between May and December 2021, participants were recruited in Switzerland through an online and offline channel. On the one hand, we posted the advertisement for the study on our private and our research group's Instagram profile (unibe_sexuellegesundheit), on the other hand, we distributed flyers at the University of Bern and in public places in the city of Bern. This recruitment method was chosen to reach a diverse group of women from the general population, rather than just women with sexual difficulties. Data were collected using the Qualtrics online questionnaire program (Qualtrics, Provo, UT). The participants were required to indicate their agreement to participate voluntarily, confirm that they were at least 18 years old, and agree that their responses would be used for research purposes on the first page of the questionnaire. If they did not agree to any of these points, they were immediately excluded from the study. After the participants had given their written informed consent, an algorithm function of the Qualtrics online questionnaire program assigned them to one of the two study conditions in a 1:1 ratio. This procedure meant that the investigators were completely blinded during the randomization and data collection. The inclusion criteria were that the participants had to be identified as (1) female, (2) over 18 years old, and (3) German speakers. All of the participants took part voluntarily. After completing the baseline questionnaire (t1), the participants were informed whether they were assigned to the IG or the WCG, with the participants in the IG receiving direct access to the online intervention and the participants in the WCG being informed that the intervention would start in 4 weeks. Four weeks later, all of participants were asked to fill out the same questionnaire again (t2), and the participants in the WCG who did so were given access to the online intervention. During the four-week intervention period, all of the participants were reminded to continue with the intervention through weekly automatic e-mails. These e-mails were sent to all of the participants regardless of whether they had used the intervention or not. The final measurement (t3) took place after another 4 weeks, following completion of the online intervention by the WCG and as a follow-up for the IG (see Figure 1 for the study design). To match each participant's responses across the three measurement points, the participants were asked to create a pseudonym, which they were asked to provide in all three questionnaires. The participants who completed all three questionnaires had the opportunity to take part in a raffle with prizes such as a book about sexual pleasure. The study was approved by the Ethics Committee of the University of Bern on 7 May 2021 (No. 2021-04-00005) and is registered on the Open Science Framework (OSF; osf.io/xbhk2, accessed on 25 January 2023).

### 2.3. Measures

The following is a description of the measurement tools and variables used for this RCT. The data collection was part of a larger project that included more measurement instruments and variables than those listed below. All information about the measures not

specified here can be found in the preregistration on the OSF (Available online: osf.io/xbhk2, accessed on 25 January 2023).

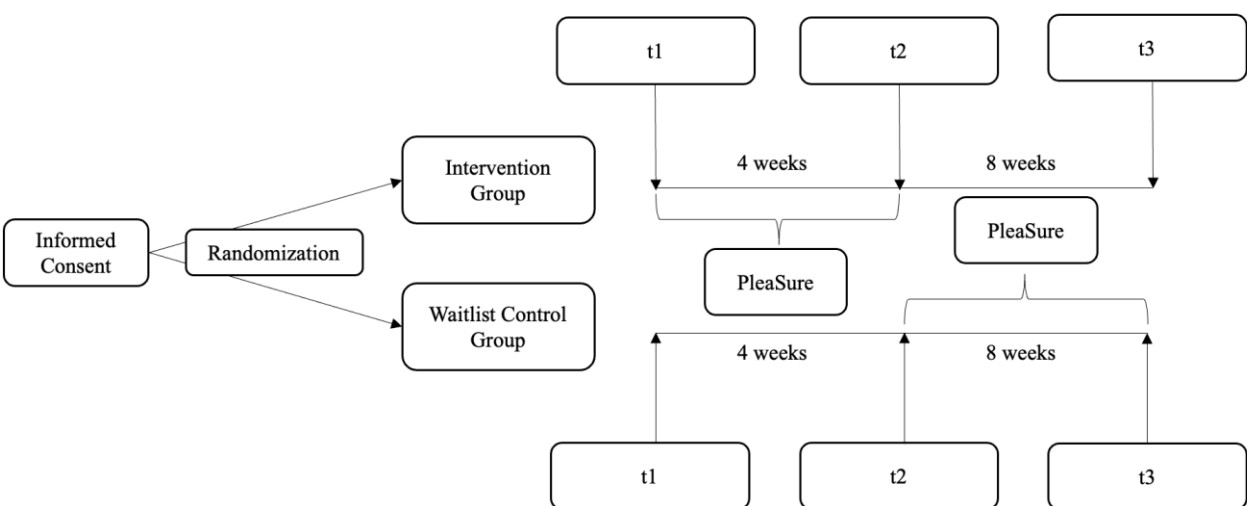

**Figure 1.** Participant flow. T1 = baseline for IG, pre-baseline for WCG; t2 = post-measurement for IG, baseline for WCG; t3 = follow-up-measurement for IG, post-measurement for WCG.

### 2.3.1. Demographic Data and Frequency of Sexual Behavior

Demographic data included information on age, gender, level of education, sexual orientation, and partnership status and were obtained in the baseline questionnaire (t1). To assess the frequency of partner sex and masturbation, two items formulated by the authors, "How often do you have partner sex on average?" and "How often do you masturbate on average?", were included in the questionnaire. Response options for each question were as follows: "I have never had partner sex or masturbated", "lLess than 1× a month", "More than 1× a month to 1× a week", "More than 1× a week to daily", AND "Several times a day".

### 2.3.2. Outcome Measure: Sexual Pleasure

The Amsterdam Sexual Pleasure Inventory (ASPI Vol. 1.0) is a self-report questionnaire comprising two parts, trait sexual pleasure and state sexual pleasure, with good psychometric properties [60]. To identify changes arising from the intervention, we assessed only the state part, which includes 30 items in six subscales. The following is a list of the ASPI's state subscales with their definitions and a sample item of each [60]:

- Sensual Pleasure: Level of pleasure experienced through sensual stimulation and its psychophysiological consequences (e.g., item 2: "Touching my erogenous zones was pleasurable."). The reliability of the five-item scale was high ($\alpha = 0.80$).
- Pleasure-Related Mastery: Level of experienced mastery in creating pleasurable sexual activities (e.g., item 6: "Shape my sex life in a way that I really enjoyed."). The reliability of the six-item scale was high ($\alpha = 0.85$).
- Pleasure-Related Validation: Level of perceived worthiness to experience positive sexual experiences and experienced self-validation during sex (e.g., item 12: "I thought it was important to live out my sexual needs."). The reliability of the three-item scale was questionable ($\alpha = 0.66$).
- Interaction Pleasure: Level of pleasure experienced during sharing pleasure and from interaction with a sexual partner (e.g., item 18: "During partner sex, we were both completely absorbed in pleasure."). The reliability of the five-item scale was excellent ($\alpha = 0.90$).
- Bonding Pleasure: Level of experienced (pleasure through) feelings of closeness, affection, safety, and security during sexual interactions (e.g., item 24: "Sex brought

me closer to my sex partner."). The reliability of the five-item scale was excellent $\alpha = 0.93$.

- General Sexual Pleasure: Level of recently experienced pleasure related to different sexual activities (i.e., items 26 and 28: "Partner sex was pleasurable." and "Masturbation was pleasurable."). The reliability of the six-item scale was high ($\alpha = 0.84$).

The subscales are calculated from the mean value and may not be added together to form a total scale. The items are rated on a 6-point Likert scale from "not at all" to "to a great extent" and refer to the experience of sexual pleasure over the previous 2 weeks. For some items, an additional response option refers to the absence of a specific event or experience and is coded "NA". Higher scores indicate higher levels of recently experienced sexual pleasure. All ASPI state items can be found in the Supplemental Material Table S1.

We chose this measurement instrument because it is the first questionnaire that tries to capture sexual pleasure holistically by covering the proposed facets of the theoretical framework for sexual pleasure [6]. Previous questionnaires capture sexual pleasure rather unidimensionally, focusing on either sensory pleasure [61,62] or pleasure during intercourse [63]. Thus, by assessing five facets of sexual pleasure sensual pleasure, pleasure-related mastery and validation, interaction pleasure and bonding pleasure), the ASPI goes beyond these questionnaires. Therefore, the ASPI enables us to gain precise insights into which facets of sexual pleasure can be improved by the intervention. Moreover, the ASPI is formulated inclusively to address all individuals including those in a relationship, singles, and sexually inactive individuals.

### 2.3.3. Compliance with and Evaluation of the Online Intervention PleaSure

The level of compliance was established with two self-formulated items assessed in the post measurement. The items related to the use of the online intervention: "In which week did you invest the most or least time in the program?" and "How often did you do the exercises on average?" The evaluation regarding the exercises was also captured after the online intervention in the postintervention measurement and assessed with the following two author-formulated questions: "Which exercise did you like? (multiple answers possible)" and "What could you benefit from the most?". Last but not least, the evaluation regarding the program in general and potential changes due to the program were assessed with the following three author-formulated open-ended questions: "Was there a key moment in your engagement with the online program and if so, what was it?"; "In the last month, what do you think was the most significant change for you that took place as a result of participating in the online program?"; and "Why was this change important to you?" The last two questions were used to obtain qualitative data about the online intervention.

### 2.4. Power Analysis

In order to ensure that our study had adequate statistical power to detect meaningful differences in our research questions (RQ1 and RQ2), we conducted two separate power analyses using G*Power (Heinrich Heine University of Düsseldorf, Düsseldorf, Germany) [64]. For our power analysis, we used a probability level of 0.05, meaning that we wanted to minimize the chance of a type I error (i.e., rejecting the null hypothesis when it is actually true) to 5%. A power of 0.8 was chosen, meaning that we wanted to have an 80% chance of detecting a true difference in the population if one existed. The effect size used in our power analysis was set to $f = 0.2$, which is considered a small-to-moderate effect size. This choice was based on the previous research in this field, which showed that the effect sizes reported in the literature are heterogeneous and can vary from small to moderate [21,65–67].

A power analysis, based on the chosen probability level, power, and effect size, determined that a sample size of $n = 52$ was needed for RQ1 and $n = 42$ was needed for RQ2. Both RQs used a repeated-measures ANOVA, with RQ1 having an additional within-between-interaction factor, requiring a larger sample size due to its complexity.

*2.5. Statistical Analysis*

Data were analyzed with IBM SPSS Statistics (SPSS) 27.0 (IBM Corp., Armonk, NY, USA). Preliminary analyses comprised a randomization check and a dropout analysis to ensure that there were no systematic differences between the groups at baseline. For the randomization check, we compared the IG to the WCG; and for the dropout analysis, the study dropouts were compared to the study remainders using independent sample *t*-tests at baseline for the demographic and descriptive variables and outcome variables. Additionally, we evaluated the comparability of the intervention and WCG in the frequency of use of the online intervention. As a prerequisite for the main ANOVAs, the outliers were identified and treated according to Tabachnick, et al. [68].

To evaluate the effectiveness of the online intervention, we computed six $2 \times 2$ mixed ANOVAs with repeated measures with the sexual pleasure subscales as dependent variables and time (t1 and t2) and condition (IG and WCG) as independent variables to analyze RQ1 and six one-way repeated measures ANOVAs with the sexual pleasure subscales as dependent variables and time (t1, t2, t3) as independent variable to analyze RQ2. We verified the assumption of normal distribution as a prerequisite for the use of these planned ANOVAs. For smaller sample sizes, where normality was not met, we used the non-parametric Friedman test. ANOVA was still used for larger sample sizes as it is known to be relatively robust to non-normality in such cases [69,70]. In addition, the effect sizes eta squared ($\eta^2$) were calculated for significant effects. Given the number of tests being run, it was important to control for the inflation of the type I error rate due to multiple testing. To address this issue, we applied the Bonferroni correction procedure to adjust the significance level for each individual test. The Bonferroni correction involves dividing the desired overall significance level ($\alpha$ set at 0.05) by the number of tests being conducted (six ANOVAs per RQ). This resulted in a adjusted significance level of 0.0083 for each individual test, ensuring that any significant results were robust and controlling for the inflation of the type I error rate [71].

## 3. Results

*3.1. Participants*

In total, $N = 963$ people were interested in participating in the study. Of this sample, $n = 661$ entered a pseudonym, which was used as the key code for merging the participants' data over all three measurement points. Four participants who indicated that they were not female (inter = 1 and male = 3) had to be excluded. Finally, 657 participants completed the baseline questionnaire fully. The mean age of the 657 participants in the final sample was $M = 31.46$ (*SD* 8.78), and they were highly educated: higher education or university: 68.9% ($n = 453$), college: 18.7% ($n = 123$), apprenticeship: 10.8% ($n = 71$), secondary school: 0.5% ($n = 3$), and others: 1.1% ($n = 7$). In total, 74.6% ($n = 490$) of the participants were heterosexual, 18.3% ($n = 120$) were bisexual, and 2.3% ($n = 15$) were homosexual. The other 4.9% ($n = 32$) of participants preferred to describe their sexual orientation differently (e.g., pansexual or heteroflex). Most participants were in a romantic relationship (65.5%, $n = 431$). Demographic and descriptive information for the IG and WCG is presented separately in Table 2.

A high dropout rate was observed between each measurement point, which was partly due to the fact that the participants' self-chosen pseudonyms could not be matched. The participant flow, which shows the dropouts between the measurement points, is displayed in Figure 2. As shown in the participant flow, most participants dropped out at the very beginning, i.e., by not completing the initial questionnaire or by not providing a pseudonym, which was a prerequisite for the data analysis and after the end of the intervention phase. Dropouts were either due to the participants not starting the questionnaire, not completing the questionnaire to the end, or not providing a pseudonym or an appropriate pseudonym. Unfortunately, it is not possible to disaggregate the ratio of reasons because we do not have other variables to relate the cases of the different time points due to data protection.

**Table 2.** Demographic and pleasure-related characteristics of the participants at baseline.

| Demographic Characteristic | IG | | | WCG | | | Significance Test | | |
|---|---|---|---|---|---|---|---|---|---|
| | *n* | M/% | SD | *n* | M/% | SD | t | df | *p* |
| Age | 340 | 31.20 | 8.79 | 317 | 31.75 | 8.77 | 0.80 | 655 | 0.425 |
| Education Level | 340 | | | 317 | | | 1.64 | 655 | 0.101 |
| Higher Education or University | 229 | 67.4 | | 224 | 70.7 | | | | |
| College | 64 | 18.8 | | 59 | 18.6 | | | | |
| Apprenticeship | 43 | 12.6 | | 28 | 8.8 | | | | |
| Secondary school | 0 | 0.0 | | 3 | 0.9 | | | | |
| Other | 4 | 1.2 | | 3 | 0.9 | | | | |
| Sexual Orientation | 340 | | | 317 | | | −0.18 | 655 | 0.857 |
| Heterosexual | 251 | 73.8 | | 239 | 75.4 | | | | |
| Bisexual | 64 | 18.8 | | 56 | 17.7 | | | | |
| Homosexual | 9 | 2.6 | | 6 | 1.9 | | | | |
| Other | 16 | 4.7 | | 16 | 5.0 | | | | |
| Relationship Status | 340 | | | 317 | | | 0.32 | 655 | 0.748 |
| Yes | 225 | 66.2 | | 206 | 65.0 | | | | |
| No | 115 | 22.8 | | 111 | 35.0 | | | | |
| Frequency of Masturbation [a] | 340 | 3.05 | 0.74 | 317 | 2.98 | 0.75 | −1.14 | 655 | 0.257 |
| Frequency of Partnersex [a] | 340 | 3.24 | 0.72 | 317 | 3.21 | 0.71 | −0.48 | 655 | 0.632 |
| Pleasure-related Variables | | | | | | | | | |
| General Sexual Pleasure [b] | 336 | 4.68 | 0.90 | 314 | 4.59 | 1.01 | −1.19 | 648 | 0.235 |
| Sensual Pleasure [b] | 337 | 4.69 | 0.93 | 312 | 4.57 | 1.07 | −1.50 | 620 | 0.134 |
| Pleasure-related Mastery [b] | 331 | 4.45 | 1.01 | 300 | 4.45 | 1.09 | 0.05 | 629 | 0.960 |
| Pleasure-related Validation [b] | 332 | 4.39 | 1.20 | 311 | 4.32 | 1.28 | −0.72 | 641 | 0.471 |
| Interaction Pleasure [b] | 237 | 4.61 | 1.10 | 206 | 4.48 | 1.13 | −1.21 | 441 | 0.225 |
| Bonding Pleasure [b] | 237 | 4.97 | 1.03 | 206 | 4.91 | 1.18 | −0.54 | 441 | 0.589 |

Note. IG = intervention group; WCG = waitlist control group. [a] range = 1–5 (1 = "I have never had couple sex/masturbated", 2 = "Less than 1× a month", 3 = "More than 1× a month to 1× a week", 4 = "More than 1× a week to daily", 5 = "Several times a day". [b] Range = 1–6.

To test the effectiveness of our intervention, we only retained participants who completed the questionnaires at preintervention and postintervention measurements for RQ1 (*n* = 235) or all three time points for RQ2 (*n* = 34).

*3.2. Randomization Check*

The groups do not differ significantly regarding their age, sexual orientation, educational level, partnership status, frequency of masturbation or partner sex, or the other outcome variables at baseline. Additionally, the groups' frequency of use of the online intervention is compared. These results confirm that randomization was successful. Values for each independent sample *t*-test are shown in Table 2.

*3.3. Dropout Analysis*

No significant difference in age, sexual orientation, educational level, partnership status, frequency of masturbation or partner sex, or the sexual pleasure outcome variables were found between the participants who remained in the study until the postintervention measurement compared to those who dropped out after the preintervention measurement. The same applies to those participants who completed all three questionnaires, with one exception: participants who remained until the follow-up measurement masturbated less at baseline (*n* = 57, *M* = 3.00, *SD* = 0.73) than those who dropped out (*n* = 600, *M* = 3.25, *SD* = 0.71; $t(655) = 2.54$, $p = 0.010$, $d = 0.2$).

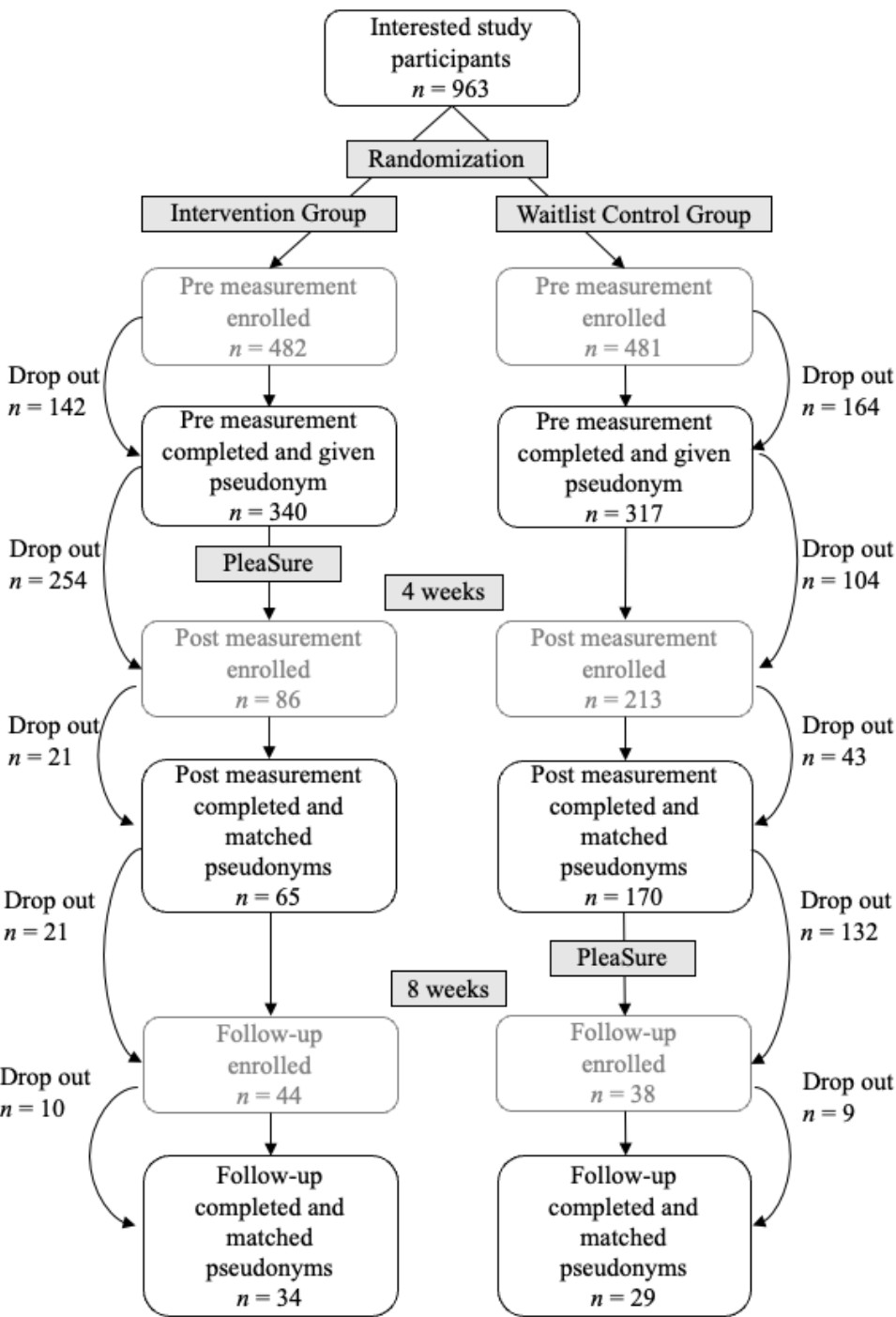

**Figure 2.** Study design and measurement time points.

### 3.4. Effects at Postintervention: Mixed ANOVA with Repeated Measures

After treating the outliers, all of the assumptions were met for conducting mixed ANOVAs with repeated measures. An increase was observed in the mean values of all of the subscales. The means and standard deviations of all of the subscales per group are shown in Table 3.

**Table 3.** Mean and standard deviation for IG and WCG at the preintervention and postintervention measurement.

| | | IG | | | | | WCG | | | |
| | | Pre | | Post | | | Pre | | Post | |
| | *n* | *M* | *SD* | *M* | *SD* | *n* | *M* | *SD* | *M* | *SD* |
|---|---|---|---|---|---|---|---|---|---|---|
| Sensual Pleasure | 57 | 4.55 | 1.03 | 4.75 | 0.88 | 148 | 4.59 | 1.03 | 4.60 | 1.04 |
| Pleasure-related Mastery | 54 | 4.26 | 1.23 | 4.60 | 1.08 | 140 | 4.50 | 1.05 | 4.44 | 1.05 |
| Pleasure-related Validation | 55 | 4.28 | 1.25 | 4.61 | 1.16 | 144 | 4.35 | 1.31 | 4.34 | 1.31 |
| Interaction Pleasure | 35 | 4.51 | 1.11 | 4.75 | 1.00 | 88 | 4.52 | 1.15 | 4.61 | 1.12 |
| Bonding Pleasure | 35 | 5.13 | 1.01 | 5.22 | 1.05 | 88 | 5.01 | 1.15 | 4.94 | 1.16 |
| General Sexual Pleasure | 56 | 4.70 | 1.01 | 4.85 | 0.82 | 148 | 4.66 | 0.93 | 4.65 | 0.97 |

Note. IG = intervention group; WCG = waitlist control group.

The mixed ANOVA for pleasure-related mastery shows a statistically significant interaction between time and group ($F(1192) = 10.77$, $p = 0.001$, $\eta^2 = 0.053$). The effects for the other subscales were not significant. The statistical values for the six $2 \times 2$ mixed ANOVAs are shown in Table 4.

**Table 4.** Results for the six $2 \times 2$ mixed ANOVAs.

| | | SS | F | df | $\eta^2$ | p |
|---|---|---|---|---|---|---|
| Sensual Pleasure | time | 0.99 | 3.02 | 1, 203 | 0.015 | 0.084 |
| | time × condition | 0.73 | 2.23 | 1, 203 | 0.011 | 0.137 |
| Pleasure-related Mastery | time | 1.60 | 5.22 | 1, 192 | 0.026 | 0.023 |
| | time × condition | 3.30 | 10.77 | 1, 192 | 0.053 | 0.001 |
| Pleasure-related Validation | time | 2.06 | 3.32 | 1, 197 | 0.017 | 0.070 |
| | time × condition | 2.37 | 3.81 | 1, 197 | 0.019 | 0.052 |
| Interaction Pleasure | time | 1.32 | 4.80 | 1, 121 | 0.038 | 0.030 |
| | time × condition | 0.26 | 0.93 | 1, 121 | 0.008 | 0.336 |
| Bonding Pleasure | time | 0.01 | 0.02 | 1, 121 | 0.000 | 0.888 |
| | time × condition | 0.30 | 0.10 | 1, 121 | 0.008 | 0.320 |
| General Sexual Pleasure | time | 0.42 | 1.39 | 1, 202 | 0.007 | 0.241 |
| | time × condition | 0.46 | 1.51 | 1, 202 | 0.007 | 0.220 |

Note. SS = sum of square numerator.

The graph in Figure 3 shows the interaction of the significant effect of pleasure-related mastery.

### 3.5. Stability of effects: One-Way ANOVAs with Repeated Measures

Six one-way repeated measures ANOVAs were performed to compare the IG's responses to the pleasure subscales before, directly after, and 4 weeks after the online intervention. The mean and standard deviation for the IG at all three measurement points are shown in Table 5.

As can be seen in Table 6, when taking into account the correction for multiple testing, none of the six one-way repeated measures ANOVAs yielded significant effects. This suggests that the online intervention did not have a notable impact on pleasure-related outcomes in the IG across the three measurement time points.

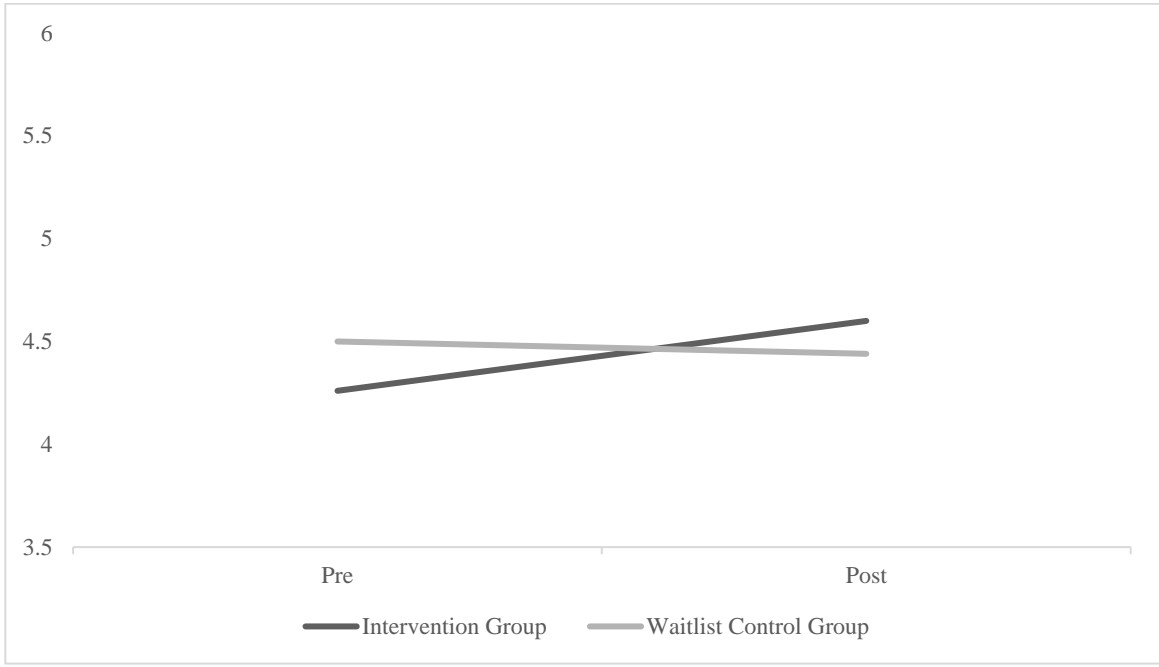

**Figure 3.** Interaction effect of pleasure-related mastery. The axis values only range from 3.5 to 6 for better visualization, but the real range is 1 to 6.

**Table 5.** Mean and standard deviation for the IG at preintervention, postintervention, and follow-up measurement.

| | n | Pre | | Post | | Follow-Up | |
|---|---|---|---|---|---|---|---|
| | | M | SD | M | SD | M | SD |
| Sensual Pleasure | 30 | 4.39 | 1.13 | 4.68 | 1.03 | 4.66 | 1.00 |
| Pleasure-related Mastery | 29 | 4.18 | 1.24 | 4.52 | 1.20 | 4.63 | 1.03 |
| Pleasure-related Validation | 29 | 4.37 | 1.23 | 4.63 | 1.09 | 4.67 | 1.21 |
| Interaction Pleasure | 19 | 4.45 | 1.08 | 4.63 | 1.00 | 4.59 | 1.07 |
| Bonding Pleasure | 19 | 5.06 | 0.96 | 5.07 | 1.16 | 5.00 | 0.95 |
| General Sexual Pleasure | 29 | 4.67 | 0.99 | 4.86 | 0.93 | 4.70 | 0.92 |

**Table 6.** Results for six one-way repeated measures ANOVA with the IG.

| | SS | F | Df | $\eta^2$ | p |
|---|---|---|---|---|---|
| Sensual Pleasure | 1.54 | 2.66 | 2, 58 | 0.084 | 0.079 |
| Pleasure-Related Mastery | 3.14 | 4.04 | 2, 56 | 0.126 | 0.033 [a] |
| Pleasure-Related Validation | 1.55 | 1.41 | 2, 56 | 0.048 | 0.252 |
| Interaction Pleasure | 0.33 | 0.65 | 2, 36 | 0.035 | 0.527 |
| Bonding Pleasure | 0.06 | 0.10 | 2, 36 | 0.006 | 0.905 |
| General Sexual Pleasure | 0.60 | 1.12 | 2, 56 | 0.038 | 0.334 |

Note. SS = sum of square numerator. [a] Huynh–Feldt correction since the sphericity assumption is not given for this subscale.

Because the normal distribution is not given for general sexual pleasure, pleasure-related mastery, pleasure-related validation, or bonding pleasure and *n* is too small (*n* < 30) for these four subscales, we additionally calculated a non-parametric test for them (Friedman test). The results did not differ from those of the ANOVAs.

### 3.6. Compliance with and Evaluation of the Online Intervention

Half of the participants (51.2%) followed our recommendations and performed the exercises at least 2–3 times a week while the others carried out the exercises less often. The majority (73.8%) stated that the combination of exercises and knowledge acquisition due to the psychoeducational elements was very helpful and better than only the exercises (9.5%), only the psychoeducational elements (14.3%), or something different (2.4%).

Among the exercises, participants favored the mindful exploration of the vulva (week 2), masturbation (week 3), and double swing (week 4) exercises. The mindful-based body scan including the genitals (week 1) and imagination of the inside of the vagina (week 4) exercises were liked by slightly fewer participants.

From the qualitative data evaluating the online intervention, we see that the participants' focus was on conscious exploration of the topic and conscious awareness of the whole body. Getting to know the body better and explicitly exploring the vulva, both in a positive way, were found helpful by many participants: "Emphasizing the importance of mindfulness and positivity towards one's own body"; "The body must learn to be touched as well." Experiencing the effects of consciously controlling breathing was also cited by some participants as a key moment: "Breathing deeply in and out helps for a more intense orgasm." In addition, the double swing exercise was often mentioned as a key moment that combines all the aspects mentioned above.

The first question about the online program in general (whether there was a key moment during engagement with the online program) was answered yes by about half of the participants (48%). In response to the two other questions about what changed for the participants as a result of the online intervention and why this change was important, a total of 66 responses were given per question. Of the 132 written answers, 20 were classified as neutral, or it was not possible to evaluate whether they indicated a positive or negative impact on the participants (for example, "More conscious perception of one's own sexuality" or "Coming to terms with one's own body"). Only one response revealed that the online program was not completed ("Unfortunately, I didn't have the energy and desire to do the program in the past weeks. But it sounded very exciting".). The remaining 111 responses, representing 86.4% of the statements, were positive, as exemplified by 18 quotes in Table 7. The participants stated that their attitude towards masturbation and the appearance of their vulva had changed for the better. In addition, some reported more self-confidence, empowerment, and that they had learned to stand up more for their own needs and become more active. There were also some participants who made the link to general life, in the sense that it improved through improved sexuality (see Table 7, quotations 15 and 16).

**Table 7.** Some answers to the questions: "In the last month, what do you think has been the most significant change for you that has taken place as a result of participating in the online program?" and "Why was this change important to you?".

| | |
|---|---|
| Quote 1 | "I have a totally new view of my sexuality. Before, it all scared me and today, I know exercises and steps to approach it. I have sex very differently than before this month". |
| Quote 2 | "At the beginning of the month, I perceived the lack of a (sexual) partner as something negative [...], now I still wish for someone with whom I could try out what I have learned here, but at the same time, I am much more in tune with myself and I am satisfied in and with my sexuality". |
| Quote 3 | "That I found my way back to more pleasurable masturbation and was ready to explore myself in a new way". |
| Quote 4 | "The relationship with my sexuality has strengthened in a positive way just by dealing with myself and the gained knowledge through the program". |
| Quote 5 | "Recognizing one's own sexual needs and seeing them as normal. Learning to control arousal and feeling that you can actively influence it during couple sex and have it in your own hands". |

**Table 7.** *Cont.*

| | |
|---|---|
| Quote 6 | "I found it easier to talk about it [sexuality]. I am more open. I use my toy differently. I try to stimulate myself in a more varied way. I try to be more aware of my body, to breathe more and to use my body differently. I would say that I have more confidence to express my needs and fears to my partner". |
| Quote 7 | "I feel much more comfortable with my genitals than before the program". |
| Quote 8 | "I experienced an orgasm for the first time". |
| Quote 9 | "I was never dissatisfied with my vulva, but I didn't pay any more attention to it than I did during masturbation. Now I'm proud of it, I know exactly how it looks when I'm not aroused, when I'm aroused, and even when I have an orgasm". |
| Quote 10 | "Knowing what my vulva really looks like and finding her beautiful has had a positive effect on masturbation and couple sexuality. I have become more confident in this regard and can generally enjoy my sexuality more". |
| Quote 11 | "Realization that I am fully functional after all". |
| Quote 12 | "A new area is opening up in my sexuality!" |
| Quote 13 | "The change has been significant for me because now instead of a feeling of lack dominating my thoughts, I feel complete and good about the way my life is right now." |
| Quote 14 | "It has brought us once again after 16 years of relationship even closer to each other and we enjoy our sexuality more than ever." |
| Quote 15 | "Now I can finally live and feel myself. (put harshly—but true)". |
| Quote 16 | "I found it frightening how little I had dealt with it before and what a big positive difference the confrontation with the vulva makes in my sexual life, but also in my normal life". |
| Quote 17 | "Because I feel more and more that I can perceive and acknowledge my needs and feelings better and better, and it is becoming easier and easier for me to show and communicate what is good for me, what I want, and where my limits are". |
| Quote 18 | "Sexuality was previously very shameful. I felt somewhat exposed and passive with regard to the speed and sequence of the build-up of arousal. I can now actively control this better". |

## 4. Discussion

This RCT is the first to empirically examine the effectiveness of an unguided online intervention for promoting sexual pleasure. The intervention, called PleaSure, uses exercises from the Sexocorporel approach to train participants to vary breathing, tension, movement, and rhythm and to provide knowledge to promote women's sexual pleasure. The main finding is that the online intervention only enhanced pleasure-related mastery; all other facets of sexual pleasure were not affected by the intervention. That the result pleasure-related mastery was promoted is not surprising, as the content of the online intervention was specifically intended to strengthen the individual skills and abilities that lead to sexual pleasure and to expand the repertoire of ways of doing so. Moreover, a focus on empowering individuals to experience more pleasure and supporting them in exploring themselves physically and genitally directly affects pleasure-related mastery.

However, the fact that all other facets of sexual pleasure remained unaffected was surprising. The interpersonal component (consisting of bonding pleasure and interaction pleasure) may not be promoted because the online intervention did not focus on partner sexuality per se and did not include the partners. For example, the online intervention did not focus on how participants stimulate partners or how partners stimulate participants; however, these are items intended to measure interaction pleasure: items 16 and 17 of the ASPI are "Stimulating my sex partner was pleasurable," and "Being stimulated by my sex partner was pleasurable." Furthermore, it is worth noting that only half of the participants in the study were partnered, which may have affected the results of the study. People in stable partnerships generally have more opportunities to practice and consolidate their newly learned knowledge and experience of sexuality, as they have a larger learning environment through regular sexual activity with their partners [72]. However, it remains

uncertain whether the partnered participants implemented the exercises within the context of their relationships, as this was not specifically assessed. Apart from that, transferring the practices to couple sex might require more routine and time that were not available in this brief intervention, even if they implemented the practices during sex with their partner. Therefore, a future version of the online intervention could put additional focus on the interpersonal domain of sexual pleasure by including partners and explicitly investigating their influence. A longer follow-up could also help to show the intended effects, because it is likely that the transfer to couple sexuality requires more time and routine. Finally, contrary to our expectations as well, no effect was found on sensual pleasure. Because some of the exercises were specifically related to self-stimulation, we expected that sensual pleasure would increase as a result of the online intervention. The qualitative data confirm an assumption of a mediating effect, as many participants cited engagement with their own vulvas as a key moment (quotes: 7, 9, 10, and 16). This indirect effect on sensual pleasure via genital self-image should be investigated further. Furthermore, we would like to emphasize here that focusing on sexual doing and experiencing may also be temporarily inhibitory to sexual pleasure because one becomes aware of one's limitations or lack of learning in sexuality.

In summary, the quantitative data of the study shows small and sobering effects on sexual pleasure. In contrast, the qualitative statements from the participants indicate that the intervention had a positive effect on overall sexual experience. Given the qualitative data, it is suggested that future studies include additional measures such as knowledge assessment, sexual satisfaction, or self-esteem to provide a more holistic understanding of the effect of the intervention. Moreover, it is important to note that the qualitative statements must be considered in the context of social desirability bias and the possibility of this influencing the results cannot be ruled out.

However, the contradictory findings of the quantitative and qualitative data raise the question of how this can come about. There are at least four possible reasons for this: first, it is possible that the psychoeducational elements and exercises of the online intervention based on Sexocorporel were not specific enough to target facets of sexual pleasure, but comprehensive enough to yield positive change in the overall sexual experience. Second, it is important to consider that the intervention may not have resulted in changes in behavior, which could explain the lack of changes in sexual pleasure [73,74]. It is indeed a limitation that behavior (e.g., if the physical exercises were internalized and applied) was not included in the study. Examining changes in behavior as an intermediate step between the intervention and changes in sexual pleasure could provide valuable insight into the mechanisms underlying any observed effects. Without this information, it is difficult to understand the relationship between the intervention and (non-)changes in sexual pleasure. Third, it is possible that there were other variables that were not covered in the study and therefore could not be controlled for contributed to the lack of changes in sexual pleasure, such as menstrual cycle, stress, or conflicts [75–77]. Having multiple measurement points and controlling for these variables would have provided more insight into the relationship between the intervention and changes in sexual pleasure. Fourth, it is possible that the participants did not actually perform the exercises as recommended, as compliance data indicated that only half of the participants adhered to the recommendation to perform the exercises at least 2–3 times per week. Since these are self-reported data, they may also not be entirely accurate, and it is possible that many participants did not follow the program as recommended. We assume that the effects could have been stronger and more evident if all participants had performed the exercises more consistently. To overcome this, future versions of such an online intervention should include more guidance, implement more motivational and reminder emails, or at least weekly surveys to ensure better control of the participants [78]. Another point to address and control this limitation is the inclusion of objective measures such as web analytics. Using web analytics would provide valuable objective data on participants' usage and engagement with the intervention. This could include tracking the number of times specific modules and exercises were accessed, as well

as the duration of time spent on each module. By analyzing these data, the researchers would have a better understanding of which exercises were most popular and effective, as well as which participants were more likely to stick to the program. Additionally, this would allow the researchers to identify any patterns or trends in participants' engagement, which could inform the design and implementation of future interventions.

*Limitations and Strengths*

Our pilot study has several limitations that should be considered. First and probably the most significant limitation of this study is the high dropout rate, which is why we would like to examine it in more detail here as the first point of our interest. This limitation can be attributed to several factors. One of the main reasons is the unguided nature of the online intervention. Studies have shown that unguided online self-help interventions typically have small effects and high dropout rates, and our study was no exception [43,79]. This is because unguided interventions rely on the participant's motivation and self-discipline to complete the program, and many participants may lose interest or become disengaged before completing the program. Another reason for the high dropout rate could be the participants' inability to reproduce their pseudonyms, which may have led to confusion and frustration for some participants, resulting in them dropping out of the study. Such dropouts must be avoided in future studies. Additionally, it is worth noting that this is a common problem in many online interventions and not specific to our pilot study. It has been reported in various studies that dropout rates can be as high as 80% in online interventions [80]. This raises the question to what extent we can trust such studies with such a dropout rate. Therefore, it is of utmost importance in future studies consider strategies to address this issue, such as providing more structure and support to increase engagement and retention or respond more to individual needs [81].

Second, the significant effect was not stable over time, which could be attributed both to the limited intervention fidelity and to the power not achieved for these analyses. Third, future studies should definitely include a follow-up for the WCG to make group comparisons and thus reach more powerful conclusions. Fourth, the sample may not be representative of the general population; the participants were highly educated (68.9% had a higher education or university degree) and showed high pleasure scores at baseline, with the latter being consistent with the distribution of pleasure in samples of the general population [82]. This might be explained by the fact that the people who are most likely to participate in such a study are already interested in sex and sensitized to the topic. However, this sampling bias arguably strengthens our finding on pleasure-related mastery, because the effect was even evident in such a sample. Future studies are necessary to investigate such brief interventions in a more heterogeneous sample, especially among people in clinical and non-clinical settings who report limited sexual pleasure or function.

Overall, in addition to these limitations and potential for improvement, this pilot study shows essential strengths. First, PleaSure is the first theory-based online intervention that tests the content of the Sexocorporel approach. Moreover, it is worth noting that while there are already other programs and apps that aim to promote sexual pleasure, our intervention is unique in that it is the first to be empirically tested. This allows for a better understanding of the intervention's effectiveness and provides a foundation for future research in this area. Since PleaSure is addressed to the general population, our intervention considers sexual pleasure as a preventative factor for sexual and public health [12]. Such interventions are still rare currently; most online interventions conducted so far involving sexual pleasure have focused primarily on at-risk or clinical subgroups [29–31,36]. The research group is also currently adapting the online intervention to men and will investigate its effectiveness in a further study. Finally, if the intervention shows a small effect in the general population, it likely has great potential to be more effective for clinical subgroups, for example, people with sexual dysfunction. People with sexual dysfunction may have more room for improvement and thus a larger potential for change than the general population. Furthermore, the lack of effects found in the general population may be due

to ceiling effects, where the majority of participants already have a high level of sexual pleasure and thus may not have as much room for improvement. Targeting a population with lower baseline levels of sexual pleasure may yield more significant results. Lastly, the online setting is in line with the current trend. In recent decades, a promising trend in sexual health interventions has emerged in which interventions are increasingly available online [35,66,67,83–87]. Furthermore, as already shown in a study on the promotion of genital self-image by Gonin-Spahni [79], the combination of knowledge acquisition and exercises were crucial for the success of the intervention. The online intervention therefore benefits from many advantages and might also be applied in combination with counseling or therapy [42,43].

## 5. Conclusions

In conclusion, this RCT is the first to empirically investigate the effectiveness of an unguided online intervention, PleaSure, for promoting sexual pleasure in women. The results of the study indicate that while the intervention was effective in increasing pleasure-related mastery, there were no significant effects on other facets of sexual pleasure. Qualitative data from the participants suggest that the intervention had a positive impact on overall sexual experience. However, the high dropout rate is a significant limitation of the study and warrants further investigation. The study provides initial evidence for the potential usefulness of the PleaSure intervention, but further improvements to the intervention's content and design are needed to better target the intervention for sexual pleasure and reduce dropout rates. Additionally, it highlights the need for more effective ways to engage participants in unguided online self-help interventions. Therefore, while the study presents some promise, more research is needed to fully understand the efficacy of the PleaSure intervention and to improve its design and engagement strategies. It should be noted that its despite limitations, the intervention may be useful for people who are motivated to engage with it and it is a low-cost option for many.

**Supplementary Materials:** The following supporting information can be downloaded at: https://www.mdpi.com/article/10.3390/sexes4010012/s1, S1: Introduction; S2: Basics; S3: Week 1; S4: Week 2; S5: Week 3; S6: Week 4; S7: Conclusion; Table S1: ASPI state items.

**Author Contributions:** Conceptualization, M.B. and S.G.-S.; data curation, M.B., L.M.B., L.A., J.V. and S.G.-S.; formal analysis, M.B. and L.M.B.; funding acquisition, S.G.-S.; investigation, M.B., L.A., J.V. and S.G.-S.; methodology, M.B., L.M.B., L.A., J.V. and S.G.-S.; project administration, M.B. and S.G.-S.; resources, M.B. and S.G.-S.; software, M.B., L.M.B., L.A., J.V. and S.G.-S.; supervision, M.B. and S.G.-S.; validation, M.B., L.M.B. and S.G.-S.; visualization, M.B.; writing—original draft, M.B.; writing—review and editing, M.B. and S.G.-S. All authors have read and agreed to the published version of the manuscript.

**Funding:** This research received no external funding.

**Institutional Review Board Statement:** The study was conducted in accordance with the Declaration of Helsinki and approved by the Ethics Committee of the University of Bern on 7 May 2021 (No. 2021-04-00005).

**Informed Consent Statement:** Informed consent was obtained from all of the subjects involved in the study.

**Data Availability Statement:** The data that support the findings of this study are available from the corresponding author, [M.B.], upon reasonable request.

**Acknowledgments:** We thank Julia Hegy and Simon Milligan for their helpful comments on the manuscript. We would also like to show our gratitude to Evelyn Vonesch for the beautiful illustrations that adorn our online intervention.

**Conflicts of Interest:** The authors declare no conflict of interest.

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
