# Peer review of "Effectiveness of an Unguided Online Intervention for Sexual Pleasure in Women: A Randomized Controlled Pilot Study"

_sexes, doi:10.3390/sexes4010012_

Round 1

Reviewer 1 Report

Congratulations for the work done, very interesting and well structured.

I think could be useful reduce Introduction and Discussion that are too extensive.

Author Response

Dear Reviewer I,

Thank you for your kind words and valuable feedback on our work. We appreciate your input and are glad to hear that you found our work to be well-structured and interesting.

Regarding the length of the Introduction, we did not make any adjustments as it was considered to be well-written by other reviewers. However, if you have specific suggestions for cuts or changes, we will gladly take them into consideration in a future revision. The Introduction was actually expanded to include information on female pleasure and additional details on pleasure and the state of the art, as requested by another reviewer, in the hopes of providing better context for the topic at hand. We hope that this additional information is accepted and understood as beneficial.

We have made significant changes to the Discussion and Conclusion sections, taking into account feedback from other reviewers. While this may not have shortened the Discussion, we believe that it has improved its rigor and relevance and hope that it meets your satisfaction as Reviewer I.

Thank you again for your valuable feedback, and we look forward to continuing to improve our work in the future.

Best regards,

Reviewer 2 Report

Dear Authors,

Manuscript contains important information for the reader.  The findings are interesting. But I have a some important suggestions in the text. The work needs major revision.

Best regards.

Reviewer 3 Report

Thank you for the opportunity to review the manuscript. The manuscript is exceptionally well-written, especially at the front end, and addresses a potentially important and interesting intervention. The study was preregistered, methods well-described, and the authors followed an RCT protocol for the study allowing them to compare treatment and control groups. The dropout rate was very high, but similar to many other online interventions of this kind. There are some clear weaknesses of the approach and the results are not very positive and thus the discussion and conclusion should be tempered. However, despite the weaknesses and difficulties encountered that are typical of these types of interventions, the study was well-developed, the manuscript carefully written, and the results of the intervention (including the difficulties with it) should be published. I have provided more detailed comments below that will hopefully further improve the manuscript.

Methods and Results:

·      This is a minor point, but it would help distinguish between the weeks in Table 1 if there was a space between each week. For example, 1st and 2nd week merge into one.

·      Partnered sex is encouraged as part of the program but being in a relationship was not an inclusion criterion. Were there differences in the usefulness of the intervention based on whether participants were in a relationship/did partnered exercises or not?

·      The initial sample size is over 10-times higher than needed based on the power simulation. Could the authors add the reasoning for this (I assume because of the high dropout rate). Also, it would be important to know during which period the data were collected and how long it took to collect the data? And when and why the authors decided to stop collecting further data?

·      I’m not clear on what means and SDs mean in relation to education level/sexual orientation/relationship status. Unless I’m misunderstanding, these numbers are meaningless and should be deleted.

·      Could it be clarified whether the dropout of 254 (and the respective drop-out rates in other group or time) in the treatment group refer to people who didn’t start the follow-up survey and the 21 people comprise of participants whose pseudonyms were not matched or were there more people within the 254 that could not be matched but did complete the second survey?

·      Was there a way to see how many people had accessed different modules (the ones who did not continue)? It would be very interesting to understand at which point participants dropped out and if they ever even engaged with the intervention. Or how much of the intervention they engaged in. The dropout rate is very high although similar to what I have also found in my own research on similar interventions.

·      Did the authors collect email addresses from the participants? Would it be possible to contact them to understand more about why the participants did not complete the intervention/did not reply at the second time point?

·      It might be useful to describe in more detail what the advertisement for the study was. It sounds like the study wasn’t really designed to study people who had difficulties sexually but could include anyone that might just have been interested in sex. This is not a problem but it’s not quite clear from the study what the intended sample was.

·      Were there significant differences in the pleasure items between participants who continued with the study and those who did not? The authors mention no differences in terms of demographics but unless I missed it, I couldn’t see whether any differences in the pleasure scale were identified.

·      This interpretation of the results is not correct: “a significant time effect (F(1,192) = 5.22, p = .023, η² = 0.026), meaning that both groups had higher values in Pleasure-Related Mastery at the postintervention measurement.” Even though the time effect is significant, it is clear that based on the means at pre- and post-intervention, the time effect is driven solely by the intervention group. This is not the same for Interaction Pleasure where both groups’ means did improve.

·      Given the authors run six ANOVAs for each subscale which each essentially test the same hypothesis (does the intervention increase pleasure), I think it would be prudent to control for multiple testing given the probability of a false positive increases a lot which each additional analysis run. Correcting for multiple testing would render all other effects non-significant apart from time*condition on pleasure-related mastery.

Discussion:

·      In the discussion, authors discuss potential reasons why only the intrapersonal not interpersonal pleasure changed. I wonder if the results are also due to the fact that only about half the participants were in a relationship (and thus would not necessarily have had an opportunity to engaged in partnered sex)? It’s also not clear whether any of the participants engaged in exercises with their partner (which were recommended as some of the exercises).

·      There are also some interpretations in the discussion section based on the significant time effect of one of the analyses which is driven only by the interaction group that should be changed accordingly (see my earlier comment on the results section).

·      One of the items of the pleasure scale mentioned in the discussion states that “during partner sex, my genitals glowed with excitement” sounds very odd in English and I wonder if this is just because of the translation from German? Genitals cannot glow (at least not in English).

·      I wouldn’t make too much of the interaction pleasure changing over time in the control group. The p-value is .03 and if the p-value is corrected for multiple testing, this effect is not significant. It’s likely a fluke given the small sample size and multiple testing.

·      How many participants responded to the qualitative questions? How many of them mentioned something negative? Could the overwhelmingly positive responses in the qualitative responses just be because only participants who liked the intervention continued with the study? Or they didn’t mention anything negative because it wasn’t asked about?

·      I’m not sure I agree that the intervention is the first that seeks to directly promote sexual pleasure as an outcome. There are several apps in this space that very much focus on pleasure, at least as one of the outcomes and later on authors themselves mention other interventions involving sexual pleasure. Maybe the authors mean interventions that have been empirically evaluated that they are aware of?

·      I would temper the conclusions. The conclusion sounds nice but is not very consistent with the results which show very mixed findings. The dropout rate is quite high and the results largely non-significant. The qualitative results show some promise and I think the intervention can be useful, and can certainly be further improved to be more useful, especially for people who are motivated to engage with such an intervention. It’s also going to be a low-cost option for many so the potential drawbacks of it are likely minimal.

·      In conclusion, I think the manuscript is very well written especially in the introduction and methods (although there are some further details and clarification which will add some details I was interested in as a reader). However, I think the manuscript would be much stronger scientifically if instead of authors trying to make the results sound better than they are (I know we all do this and have to sell our papers to some degree so it’s difficult to find a balance), they engaged more with a discussion around what were things that did work (I think there are some very strong points in the results section that could be elaborated on further from the qualitative and overall feasibility questions), why the dropout rate was so high and can we trust the results given the dropout, what would have made the intervention better, what are the implications of so many people being interested but not actually engaging with online interventions (in general), what do the results say about feasibility of online interventions etc.? I wouldn’t expect the authors to engage with all of these questions in the discussion, but I would like the discussion to be more directed towards a discussion of the limitations than overinterpreting results that we don’t really know how reliable they are.

Finally, I want to reiterate that I don’t think the (lack of positive) results should in any way weigh in on whether the study should be published or not. I think the paper is very important and should be published and we should be having a discussion around online interventions and how we can move towards making them more engaging and useful for people and we can’t do that if we only publish the few that by some miracle managed to keep most of the participants.
